# Vertically optimized phase separation with improved exciton diffusion enables efficient organic solar cells with thick active layers

Yunhao Cai[1], Qian Li[2], Guanyu Lu[3], Hwa Sook Ryu[4], Yun Li[1], Hui Jin[5], Zhihao Chen[6], Zheng Tang [5], Guanghao Lu [3✉], Xiaotao Hao[6], Han Young Woo [4], Chunfeng Zhang [2✉] & Yanming Sun [1,7✉]

The development of organic solar cells (OSCs) with thick active layers is of crucial importance for the roll-to-roll printing of large-area solar panels. Unfortunately, increasing the active layer thickness usually results in a significant reduction in efficiency. Herein, we fabricated efficient thick-film OSCs with an active layer consisting of one polymer donor and two non-fullerene acceptors. The two acceptors were found to possess enlarged exciton diffusion length in the mixed phase, which is beneficial to exciton generation and dissociation. Additionally, layer by layer approach was employed to optimize the vertical phase separation. Benefiting from the synergetic effects of enlarged exciton diffusion length and graded vertical phase separation, an efficiency of 17.31% (certified value of 16.9%) is obtained for the 300 nm-thick OSC, with a short-circuit current density of 28.36 mA cm$^{-2}$, and a high fill factor of 73.0%. Moreover, the device with an active layer thickness of 500 nm also shows an efficiency of 15.21%. This work provides valuable insights into the fabrication of OSCs with thick active layers.

[1] School of Chemistry, Beihang University, 100191 Beijing, P. R. China. [2] National Laboratory of Solid State Microstructures, School of Physics, and Collaborative Innovation Center for Advanced Microstructures, Nanjing University, Nanjing 210093, P. R. China. [3] Frontier Institute of Science and Technology, Xi'an Jiaotong University, Xi'an 710054, P. R. China. [4] Department of Chemistry, College of Science, KU-KIST Graduate School of Converging Science and Technology, Korea University, Seoul 136-713, Republic of Korea. [5] State Key Laboratory for Modification of Chemical Fibers and Polymer Materials, Center for Advanced Low-dimension Materials, College of Materials Science and Engineering, Donghua University, Shanghai 201620, P. R. China. [6] School of Physics State Key Laboratory of Crystal Materials, Shandong University, Jinan 250100, P. R. China. [7] Beijing Advanced Innovation Center for Biomedical Engineering, 100191 Beijing, China. ✉email: guanghao.lu@mail.xjtu.edu.cn; cfzhang@nju.edu.cn; sunym@buaa.edu.cn

Organic solar cells (OSCs) have been the focus of a burgeoning research effort for their appealing advantages of solution processability, cost-effectiveness and mechanical flexibility[1–5]. In the past few years, the rapid developments of fused-ring non-fullerene acceptors (NFAs) have led to continuous improvements in power conversion efficiencies (PCEs) of OSCs, and the champion devices have yielded high PCEs over 18%[6–15]. Such significant achievements demonstrate that OSCs are close to their large-scale commercialization. However, state-of-the-art OSCs with high efficiencies are typically achieved with an optimal active layer thickness around 100 nm, which is unsuitable for the upscaling aim since it's difficult to produce uniform and defect-free films upon industrial high throughput manufacturing[16–20]. Consequently, employing thicker active layers over several hundred nanometers that can largely expand the printing process window, is essential to meet the requirement for large-scale fabrication. However, increasing the active layer thickness usually results in decreased PCEs[21–24]. Therefore, the fabrication of efficient thick-film OSCs is considered as a critical challenge.

Currently, most of OSCs are constructed with a bulk heterojunction (BHJ) architecture, in which an electron donor (D) and an electron acceptor (A) are mixed to form a bicontinuous interpenetrating D/A network[25–29]. After light absorption of incident light by the active layer, tightly bound electron-hole pairs (known as excitons) are generated and the excitons need to travel to the D/A interfaces to dissociate into charge carriers[30,31]. Accordingly, the exciton diffusion length ($L_D$) is a key parameter in OSCs, as it determines the amounts of excitons that reach D/A interfaces[32–34]. However, there is a sharp mismatch between the $L_D$ (typically around 10 nm) and the scale of phase separation in BHJ OSCs. An enlarged $L_D$ is therefore highly desired, especially for thick-film OSCs, which allows a significant fraction of excitons cross a longer distance to diffuse to D/A interfaces, contributing to a higher photocurrent generation[35,36]. On the other hand, the charge recombination is another main factor that limits the PCE of thick-film OSCs. The low and unbalanced charge transport, the increased trap state density, and the space charge accumulation can all lead to severe charge recombination in thick-film devices[19,20,37,38]. Indeed, for thick-film fullerene-based OSCs, the highest efficiency is only 11.3% with the active layer thickness of 280 nm due to the unbalanced hole and electron mobilities in the device[19,20]. To solve this problem, the most effective strategy is to optimize the thick active layer morphology, however, it remains quite difficult. The thickness of an active layer strongly depends on the viscosity and concentration of the D/A solution. A high concentration of D/A solution is typically utilized to produce thick films. In comparison with thin films, thick films typically undergo a longer phase evolution time during solvent evaporation and larger space (in the vertical direction) is available for phase transition, both of which could induce more significant phase separation. In this perspective, forming a well-defined vertical phase separation, which can enhance D/A interface area, facilitate charge transport, and suppress charge recombination, is a prerequisite for fabricating thick-film OSCs.

Herein, we demonstrated efficient thick-film OSCs fabricated with a combination of PM6 donor, BTP-eC9 and L8-BO-F acceptors. The two acceptors not only have complementary absorption, but also show enlarged exciton diffusion length in the mixed phase, which could facilitate exciton generation and dissociation upon increasing the active layer thickness. The efficient energy transfer between BTP-eC9 and L8-BO-F, and the increased crystallinity in their blend help to enhance the exciton diffusion. PM6:BTP-eC9:L8-BO-F OSCs with thick active layers realize obviously higher PCEs than the binary PM6:BTP-eC9 devices. We subsequently used layer by layer (LBL) processing strategy to further optimize the vertical phase separation of PM6:BTP-eC9:L8-BO-F films. As a result, the 300 nm-thick device yields a PCE of 17.31% (certified value of 16.9%), with a short-circuit current density ($J_{sc}$) of 28.36 mA cm$^{-2}$, an open-circuit voltage ($V_{oc}$) of 0.836 V, and a high fill factor (FF) of 73.0%. When the thickness of active layer reaches 500 nm, an efficiency of 15.21% is also obtained. Systematic studies reveal that the long-lived charge carriers, effective charge transport, suppressed charge recombination and graded vertical phase separation account for the better photovoltaic performance achieved in the LBL-processed thick-film OSCs. In light of this strategy, another two different acceptors, which possesses improved exciton diffusion length in the mixed film, have been selected to fabricate thick-film OSCs. As a result, 300 nm-thick PM6:Y6:Y6-F OSCs also produce a PCE of 15.42%. Our work highlights the role of exciton diffusion length and graded vertical phase separation in realizing thick-film OSCs with better photovoltaic performance.

## Results

**Photovoltaic performance.** Figure 1a shows the chemical structures of the donor and acceptor materials used in this work. For clarity, hereby we take PM6:BTP-eC9:L8-BO-F as a representative system to reveal the underlying mechanism for the high photovoltaic performance achieved in the thick-film devices. PM6, BTP-eC9 and L8-BO-F possess complementary absorption and L8-BO-F and BTP-eC9 can form a homogeneous mixed phase, which is beneficial to enhance the molecular packing of the materials[12,39] (Supplementary Fig. 1). In this work, OSCs with different active layer thicknesses were fabricated by adopting conventional device architecture of ITO/PEDOT/active layer/PNDIT-F3N/Ag. It should be noted that the active layer thicknesses are actually around 120, 300 and 500 nm. Instead, we use 120, 300, and 500 nm to simplify the description. A PCE of 18.46% is achieved for the 120 nm-thick ternary device. Then we increased the active layer thickness to 300 nm. It was found that the PCEs of the ternary thick-film OSCs are significantly higher than their binary counterparts. In details, the PM6:BTP-eC9:L8-BO-F device yields a PCE of 16.92%, with a high $J_{sc}$ of 28.27 mA cm$^{-2}$, a $V_{oc}$ of 0.837 V, and an FF of 71.5%, while the host PM6:BTP-eC9 device only exhibits a PCE of 15.62%, with a $J_{sc}$ of 27.64 mA cm$^{-2}$, a $V_{oc}$ of 0.820 V, and an FF of 68.9%. When further increasing the thickness to 500 nm, the ternary devices also exhibits superior PCEs than the binary devices. It can be seen that in relative to the thin-film OSCs, the role of the third component is more prominent when using the same material combinations to fabricate thick-film devices.

LBL processing method was further adopted to optimize the vertical phase separation of active layers. Encouragingly, the LBL-processed PM6:BTP-eC9:L8-BO-F thick device produces a PCE of 17.31% (certified as 16.9% in the National Institute of Metrology in China, Supplementary Fig. 2) at 300 nm active layer thickness, with a $J_{sc}$ of 28.36 mA cm$^{-2}$, a $V_{oc}$ of 0.836 V, and an FF of 73.0%. At the thickness of 500 nm, the device also yields a satisfying PCE of 15.21%, with a $J_{sc}$ of 27.49 mA cm$^{-2}$, a $V_{oc}$ of 0.835 V, and an FF of 66.4%. The efficiencies of 17.31% and 15.21% outperform the PCEs reported for 300 nm- and 500 nm-thick OSCs in the literature thus far (Fig. 1b and Supplementary Table 1). The representative current-density versus voltage ($J$–$V$ under simulated AM 1.5 G illumination at 100 mW cm$^{-2}$) characteristics of the binary and ternary devices with different thicknesses are shown in Fig. 1c and Supplementary Fig. 3, and the corresponding external quantum efficiency (EQE) spectra are plotted in Fig. 1d and Supplementary Fig. 4. The EQE profiles of these blends are similar, while the ternary blend mainly raises the EQE from the 350–700 nm, and the $J_{sc}$s calculated from the EQE profiles agree well with those achieved from the $J$–$V$ measurements within 2% mismatch. The detailed photovoltaic parameters

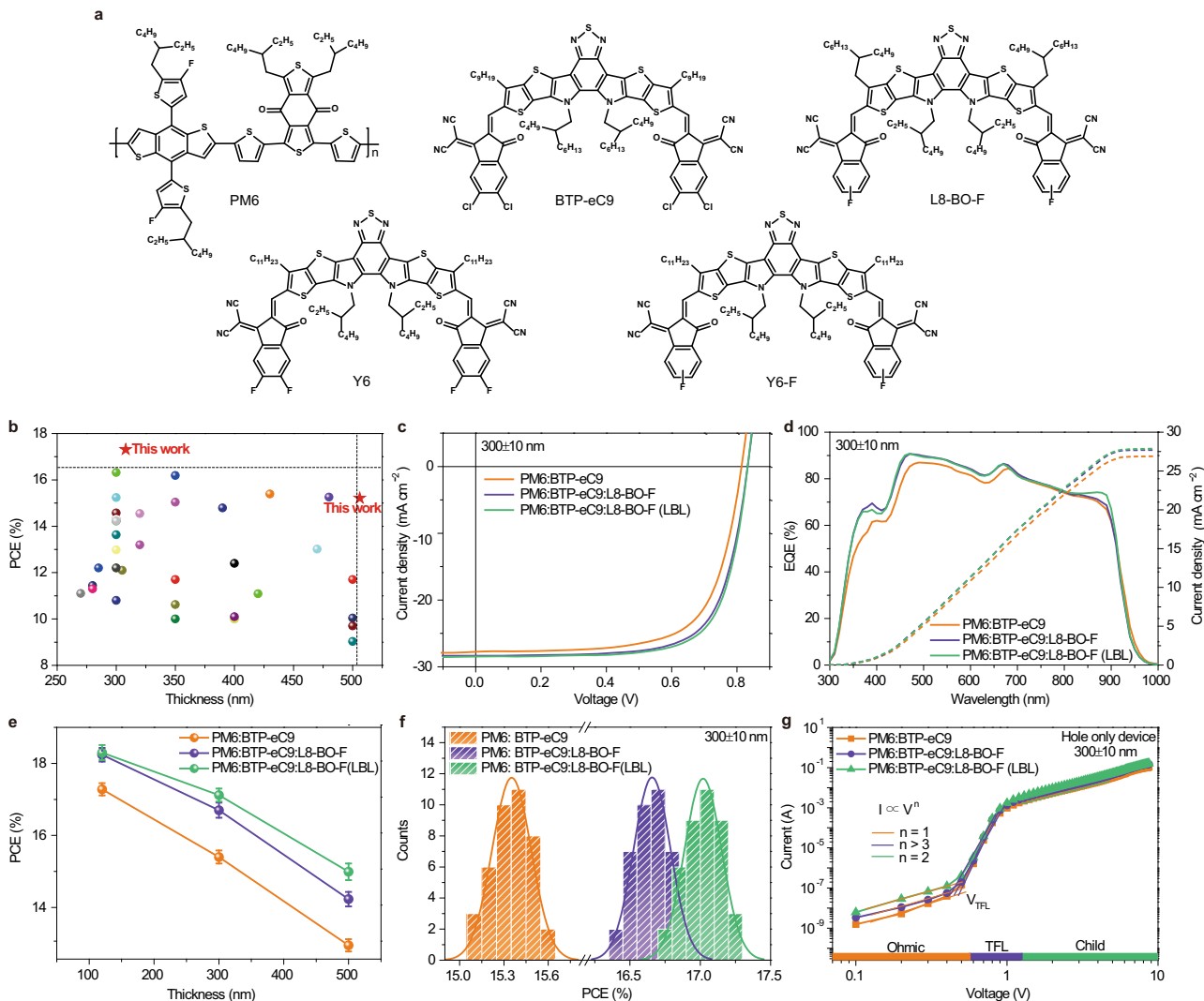

**Fig. 1 Molecular structures, photovoltaic performance and mobilities. a** Molecular structures of donors and acceptors used in this work. **b** Plots of the PCE versus active layer thickness for different systems. **c** Current–voltage (*J*–*V*) characteristics of the PM6:BTP-eC9 and PM6:BTP-eC9:L8-BO-F devices with 300 nm active layer thickness under simulated AM 1.5 G illumination at 100 mW cm$^{-2}$. **d** EQE spectra (solid lines) and integrated $J_{sc}$s (dashed lines) of the binary and ternary devices. **e** Dependence of PCEs on the active layer thicknesses (the error bars represent the standard deviations). **f** Histogram of the PCE measurements for 40 devices based on the representative binary and ternary blends with 300 nm thickness. **g** *I-V* traces of the different blends with 300 nm thickness exhibiting three different regimes, marked with Ohmic, TFL, and Child, respectively.

of these OSCs are summarized in Table 1. Figure 1e provides PCE comparisons of the binary, ternary and LBL-processed ternary OSCs with different thicknesses. It was observed that with the increase of the active layer thickness, the PCEs of LBL-processed OSCs decrease more slowly, as compared to other two blends. Besides, the evolvements of the $V_{oc}$, $J_{sc}$ and FF are collected in Supplementary Fig. 5. Figure 1f presents the efficiency distribution histogram for 40 independently measured 300 nm-thick binary and ternary devices, showing that the average efficiency of the LBL ternary devices are all higher than those of the conventional ternary counterparts, especially at thick active layer thicknesses. The LBL-processed binary devices also show the same trend with the ternary ones, indicating the general application of LBL approach in further improving the efficiency of thick-film OSCs (Supplementary Fig. 6 and Table 2). Moreover, large-area device with 1 cm$^2$ active area was fabricated. Representative *J*–*V* curve of the device and the corresponding EQE spectrum are shown in Supplementary Fig. 7. The large-area device yields a PCE of 16.01%, with a $J_{sc}$ of 28.38 mA cm$^{-2}$, a $V_{oc}$ of 0.838 V, and an FF of 67.3%. Additionally, photostability of the

thin and thick devices were measured under continuous one sun illumination in air. As shown in Supplementary Fig. 8, 120 nm-, 300 nm- and LBL-processed 300 nm-thick ternary devices reached their $T_{80}$ (the time required to reach 80% of initial efficiency) at 127, 138, and 152 h, respectively. The results indicate that the thick devices exhibit slightly improved photostability than the thin devices.

The charge carrier mobility has been identified as a pivotal factor in thick-film OSCs. The hole mobility ($\mu_h$) and electron mobility ($\mu_e$) of the active layers with different thicknesses were measured by the space charge limited current (SCLC) method (Fig. 1g and Supplementary Figs. 9 and 10), and the calculated mobilities are shown in Table 2. The device configuration of ITO/PEDOT:PSS/active layer/MoO$_3$/Al and ITO/ZnO/active layer/PNDIT-F3N/Al were used to measure the $\mu_h$ and $\mu_e$, respectively. With the increase of active layer thickness, the $\mu_h$ and $\mu_e$ of the binary and ternary devices are both improved, while the hole mobility increases more obviously (Supplementary Fig. 11). The $\mu_h$ and $\mu_e$ of LBL-processed ternary devices with 300 nm active layer thickness are $12.33 \times 10^{-4}$ and $9.04 \times 10^{-4}$ cm$^2$ V$^{-1}$ s$^{-1}$,

**Table 1 Summary of photovoltaic parameters of the PM6:BTP-eC9 and PM6:BTP-eC9:L8-BO-F devices with different active layer thicknesses.**

| Active layer | Thickness(nm) | $V_{oc}$ (V) | $J_{sc}$ (mA cm$^{-2}$) | FF (%) | PCE[a] (%) |
|---|---|---|---|---|---|
| PM6:BTP-eC9 | 120 ± 10 | 0.840 (0.839 ± 0.002) | 26.61 (26.30 ± 0.30) | 78.2 (77.5 ± 0.6) | 17.47 (17.28 ± 0.17) |
| | 300 ± 10 | 0.820 (0.819 ± 0.002) | 27.64 (27.35 ± 0.28) | 68.9 (68.2 ± 0.6) | 15.62 (15.40 ± 0.18) |
| | 500 ± 10 | 0.818 (0.817 ± 0.002) | 26.13 (25.81 ± 0.31) | 61.5 (61.1 ± 0.5) | 13.14 (12.95 ± 0.17) |
| PM6:BTP-eC9:L8-BO-F | 120 ± 10 | 0.853 (0.853 ± 0.001) | 27.25 (26.91 ± 0.32) | 79.4 (79.0 ± 0.5) | 18.46 (18.24 ± 0.19) |
| | 300 ± 10 | 0.837 (0.836 ± 0.002) | 28.27 (28.01 ± 0.25) | 71.5 (71.0 ± 0.6) | 16.92 (16.70 ± 0.21) |
| | 500 ± 10 | 0.836 (0.835 ± 0.001) | 27.40 (26.98 ± 0.33) | 63.1 (62.4 ± 0.6) | 14.45 (14.23 ± 0.20) |
| PM6:BTP-eC9:L8-BO-F (LBL) | 120 ± 10 | 0.852 (0.851 ± 0.002) | 27.26 (26.96 ± 0.30) | 79.8 (79.2 ± 0.6) | 18.53 (18.30 ± 0.21) |
| | 300 ± 10 | 0.836 (0.836 ± 0.001) | 28.36 (28.08 ± 0.27) | 73.0 (72.5 ± 0.5) | 17.31 (17.12 ± 0.18) |
| | 500 ± 10 | 0.835 (0.834 ± 0.002) | 27.49 (27.10 ± 0.37) | 66.4 (65.8 ± 0.5) | 15.21 (14.99 ± 0.23) |
| PM6:BTP-eC9:L8-BO-F (LBL) | 300 ± 10 | 0.830 | 28.14 | 72.5 | 16.9[b] |

[a]Average values with standard deviation were obtained from 40 devices.
[b]Certified by National Institute of Metrology, China.

**Table 2 Hole mobility and electron mobility of PM6:BTP-eC9 and PM6:BTP-eC9:L8-BO-F devices with different thicknesses.**

| Active layer | Thickness (nm) | Hole mobility [a] (×10$^{-4}$ cm$^2$ V$^{-1}$ s$^{-1}$) | Electron mobility (×10$^{-4}$ cm$^2$ V$^{-1}$ s$^{-1}$) | $\mu_h/\mu_e$ |
|---|---|---|---|---|
| PM6:BTP-eC9 | 120 ± 10 | 7.19 (6.96 ± 0.25) | 5.38 (5.15 ± 0.25) | 1.34 |
| | 300 ± 10 | 9.40 (9.19 ± 0.23) | 6.26 (6.01 ± 0.26) | 1.50 |
| | 500 ± 10 | 11.10 (10.83 ± 0.28) | 6.35 (6.13 ± 0.24) | 1.75 |
| PM6:BTP-eC9:L8-BO-F | 120 ± 10 | 8.33 (8.06 ± 0.26) | 6.67 (6.39 ± 0.26) | 1.26 |
| | 300 ± 10 | 11.85 (11.59 ± 0.24) | 8.51 (8.33 ± 0.20) | 1.39 |
| | 500 ± 10 | 13.54 (13.33 ± 0.20) | 8.55 (8.30 ± 0.22) | 1.58 |
| PM6:BTP-eC9:L8-BO-F (LBL) | 120 ± 10 | 8.50 (8.22 ± 0.27) | 6.75 (6.50 ± 0.27) | 1.26 |
| | 300 ± 10 | 12.33 (12.18 ± 0.25) | 9.04 (8.82 ± 0.20) | 1.36 |
| | 500 ± 10 | 14.00 (13.80 ± 0.23) | 9.10 (8.91 ± 0.21) | 1.54 |

[a]Average values with standard deviation were obtained from 20 devices.

respectively, with a $\mu_h/\mu_e$ ratio of 1.36. In addition, it has been recognized that traps usually act as non-radiative recombination centers, and the density of the trap state ($n_t$) closely related to the carrier diffusion, transport, and photovoltaic performance of OSCs, particularly in thick-film devices[40]. As shown in Fig. 1g, there are three obvious regions in $J$–$V$ curves of the hole only devices. At low voltage, the ohmic region starts, which is determined by the linear $J$–$V$ relation (orange region). When the bias voltage increases, it enters the trap-filled limited region (purple region), which can be deduced by the rapid nonlinear rise (set in at $V_{TFL}$) of injected current. The density of trap states can be calculated by the equation of $V_{TFL} = \frac{en_t L^2}{2\varepsilon\varepsilon_0}$, where $\varepsilon$ is the relative dielectric constant, $L$ is the thickness of active layer, and $\varepsilon_0$ is the vacuum permittivity[41]. The calculated average densities of trap states for 300 nm- and LBL-processed 300 nm-thick ternary devices are $1.50 \times 10^{15}$, and $1.42 \times 10^{15}$ cm$^{-3}$, respectively. The lower density of trap states in the LBL-processed device contributes to the reduced trap-assisted recombination and longer carrier lifetime in the corresponding OSCs.

The charge recombination behaviors inside the devices were then examined by measuring the dependence of $V_{oc}$ on the light intensity ($P_{light}$, Supplementary Fig. 12). When bimolecular recombination is the dominant mechanism, the slope of $V_{oc}$ versus the natural logarithm of the $P_{light}$ is equal to $k$T/q ($k$ is the Boltzmann constant, T is Kelvin temperature, and $q$ is the elementary charge)[42]. At the active layer thickness of 300 nm, the binary, ternary and LBL-processed ternary devices show slope values of 1.37, 1.32 and 1.27 $k$T/q, respectively. The 120 nm- and

500 nm-thick binary OSCs also show larger slope values than the ternary devices, which is suggestive of that L8-BO-F indeed leads to reduced trap-assisted recombination in the ternary devices. Intriguingly, the LBL-processed ternary devices show the relatively smaller slope values than the ternary devices at the same active layer thickness, suggesting that the LBL processing approach is helpful in reducing the density of trap states in the active layer and leads to reduced trap-assisted recombination. This finding also agrees well with the SCLC results, in which the LBL-processed ternary devices show the lowest density of trap states.

To investigate the charge generation and exciton dissociation behavior of OSCs with different film thicknesses, the photo-current density ($J_{ph}$) versus effective voltage ($V_{eff}$) characteristics were measured and the corresponding plots are given in Supplementary Fig. 13. $J_{ph}$ is denoted as $J_{ph} = J_L - J_D$, where $J_L$ and $J_D$ represent the current densities that are illuminated and in the dark, respectively. $V_{eff}$ is obtained by $V_{eff} = V_0 - V_a$, where $V_0$ is the voltage at which $J_{ph}$ is zero and $V_a$ is the applied bias voltage[43]. The exciton dissociation probabilities ($P_{diss}$s) calculated by normalizing $J_{ph}$ with respect to $J_{sat}$ ($J_{ph}/J_{sat}$) under the short-circuit condition was estimated to be 97.1%, 98.3% and 98.2% for 120 nm-thick binary, ternary and LBL-processed ternary devices, respectively. The $P_{diss}$s are 95.6%, 96.3% and 97.0%, for 300 nm-thick binary, ternary and LBL-processed ternary devices, respectively. These results indicate that the existence of the third component can facilitate the charge generation and exciton dissociation inside the OSCs. It was also realized that LBL strategy enables more efficient charge generation and exciton dissociation

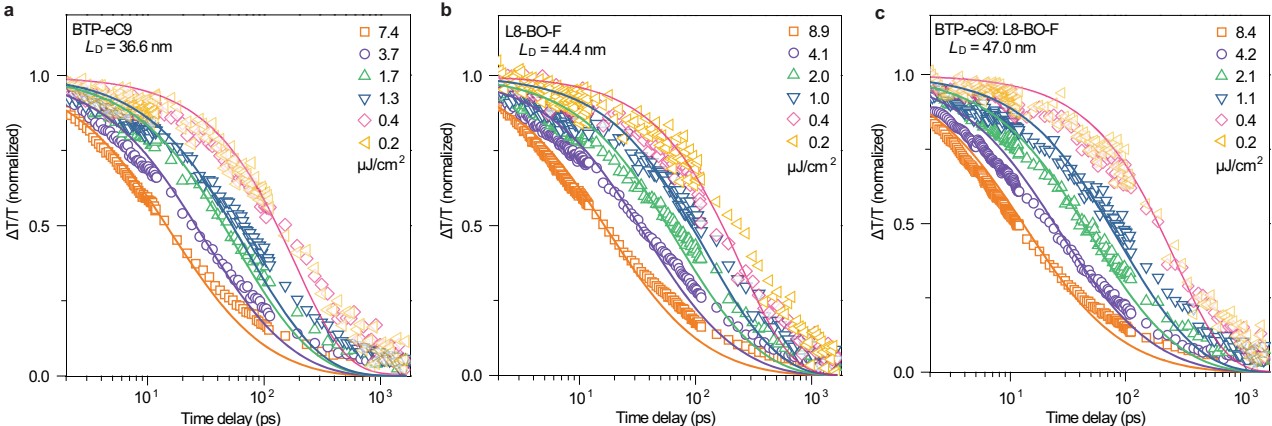

**Fig. 2 Exciton diffusion length measurements.** The dynamics of the singlet excitons measured with the 670 nm pump excitation at different densities in films of (**a**) BTP-eC9 (850 nm), (**b**) L8-BO-F (814 nm), (**c**) BTP-eC9:L8-BO-F (850 nm).The fluence-dependent singlet exciton decays are fitted to the exciton annihilation model (Eq. (1)).

in thick-film devices, as compared with conventional binary and ternary devices.

**Exciton diffusion length measurements.** From the photovoltaic performance, we found that the effect of the third component is actually more evident when fabricating thick-film OSCs. To investigate the underlying mechanism for such results, we used pump fluence-dependent transient absorption (TA) spectroscopy to measure the $L_D$, which is vitally important for the thick OSCs in the mixed acceptor films. In general, the diffusion length of excitons can be evaluated using $L_D = (D\tau)^{1/2}$, where $\tau$ is the exciton lifetime and $D$ is the diffusion constant. $D$ is given by $D = \alpha/(8\pi R)$ (three-dimensional diffusion model), where $\alpha$ is the bimolecular exciton annihilation rate constant, $R$ is the annihilation radius of excitons. The values of $\alpha$ and $\tau$ can be estimated from the experimental data of exciton decay dynamics considering the exciton-exciton annihilation (EEA). The exciton decay dynamics characterized by TA spectroscopy can be described approximately as[32,44–46]:

$$N(t) = \frac{N(0)e^{-\kappa t}}{1 + \frac{\alpha}{2\kappa}N(0)[1 - e^{-\kappa t}]} \quad (1)$$

where $\kappa = 1/\tau$ is the intrinsic exciton decay rate constant, and $N(t)$ is the exciton density as a function of time delay after excitation. Figure 2 shows the fluence-dependent decay traces of photon-induced excitons in BTP-eC9:L8-BO-F, BTP-eC9 and L8-BO-F films, respectively. The decays of the excitons are strongly dependent on the excitation intensity. According to Eq. (1), the parameters $\kappa$ and $\alpha$ are obtained with the global fitting algorithm as summarized in Supplementary Table 3. Taking the literature value of the annihilation radius ($R = 1$ nm)[32,44], the calculated value of $L_D$ increases to 47 nm in the blend from 36.6 nm in the pristine BTP-eC9 film, and 44.4 nm in the pristine L8-BO-F film, implying that the exciton diffusion is promoted in the BTP-eC9:L8-BO-F mixed phase in comparison with the neat acceptor, which is beneficial for optimizing thick-film device where the excitons are supposed to diffuse over a longer distance to reach the D/A interface. These results are also consistent well with the observed trend in device performance. The ternary OSCs exhibit better exciton dissociation and higher photocurrent density than the binary devices.

Steady-state photoluminescence (PL) and time-resolved photoluminescence (TRPL) measurements have been performed to investigate the underlying reason for the improved exciton

diffusion. It can be seen that the PL spectrum of L8-BO-F has largely overlapped the absorption of BTP-eC9 (Supplementary Fig. 14). Moreover, the emission of L8-BO-F was found to diminish, meanwhile the emission of BTP-eC9 increases upon blending with L8-BO-F. TRPL spectra shows that after adding the L8-BO-F to the host BTP-eC9 acceptor, the average fluorescence lifetime ($\tau$) is improved to 824 ps at 900 nm, much longer than that (554 ps at 900 nm) of the neat BTP-eC9 film (Supplementary Fig. 14c). To study the charge transfer behavior, OSCs based on BTP-eC9, L8-BO-F and BTP-eC9:L8-BO-F films were fabricated. As shown in Supplementary Fig. 14d, the BTP-eC9:L8-BO-F device yields a $J_{sc}$ of 0.11 mA cm$^{-2}$, which is smaller than that (0.28 mA cm$^{-2}$) of the OSC based on neat BTP-eC9, indicating negligible charge transfer between BTP-eC9 and L8-BO-F. Overall, these results suggest that the fast Förster energy transfer of excitons from L8-BO-F to BTP-eC9 occurs in their blend. Förster energy transfer can facilitate exciton hopping among conjugated segments in organic semiconductors[46]. The high crystalline order can favor the formation of longer conjugated segments and reduce the energetic disorder, enhancing Förster-mediated exciton diffusion[47]. As shown in Supplementary Fig. 15, the crystallinity of neat BTP-eC9, L8-BO-F and their blend has been investigated by grazing incidence wide-angle X-ray scattering (GIWAXS). Increased π-π stacking peak intensity with a larger crystalline correlation length (CCL) of 19.17 Å in the out-of-plane (OOP) direction in the blend film was observed compared to the neat L8-BO-F and BTP-eC9 (Supplementary Table 4). The results suggest that the addition of L8-BO-F into BTP-eC9 could increase the molecular order and enhance the exciton diffusion.

**Morphology investigation.** It has been known that the vertical phase separation of the donor and the acceptor materials in OSCs is of significant importance in influencing the exciton dissociation and charge transport. Here, we employed the film-depth-dependent light absorption spectroscopy (FLAS) to investigate the vertical phase segregation of the 300 nm-thick films. The FLAS was in-situ measured upon incrementally etching by a soft plasma, as previously reported[48–50]. The FLAS could be utilized to investigate the composition distribution along film-depth direction. The FLAS results show that the absorption peak in the wavelength range of 800–850 nm of the BTP-eC9 varies with film-depth (Fig. 3a–c), implying that the LUMO level (electron transport level) of BTP-eC9 depends on film-depth, which

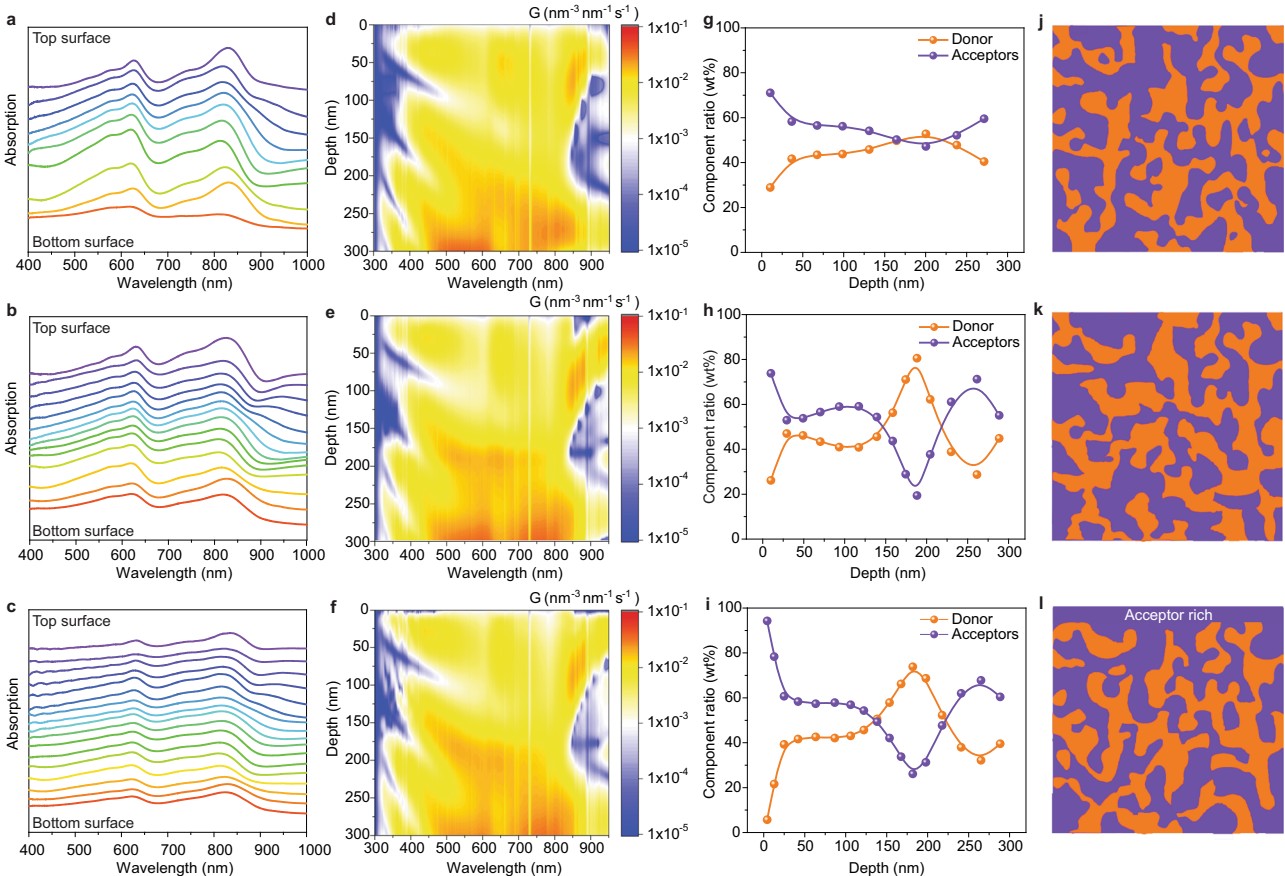

**Fig. 3 Film-depth-dependent phase segregation and exciton generation contours. a–c** Film-depth-dependent light absorption spectra of 300 nm-thick (**a**) PM6:L8-BO-F blend, **b** PM6:BTP-eC9:L8-BO-F blend, and (**c**) LBL-processed PM6:BTP-eC9:L8-BO-F blend. The spectra are vertically shifted for clarity, to show the optical properties at different film-depth. **d–f** Exciton generation contours of 300 nm-thick (**d**) PM6:L8-BO-F blend, **e** PM6:BTP-eC9:L8-BO-F blend, and (**f**) LBL-processed PM6:BTP-eC9:L8-BO-F film, as simulated from spectra of (**a–c**) in combination with optical transfer-matrix approach. The "noise"-like vertical lines are due to features of the AM 1.5G solar spectra. 0 nm and 300 nm represent active layer/PNDIT-F3N and PEDOT:PSS/active layer interfaces, respectively. The incident light is from the bottom side. **g–i** Film-depth-dependent composition profiles as extracted from (**a–c**). The scatters are from experimental measurements and the lines are guides to eyes. **j–l** Schematic illustration of the vertical phase separation of different blends: **j** PM6:L8-BO-F blend, **k** PM6:BTP-eC9:L8-BO-F blend, and (**l**) LBL-processed PM6:BTP-eC9:L8-BO-F blend with 300 nm thickness.

inevitably induces some low-energy-level regions (or traps) and thus deteriorates electron transport[49]. Therefore, the introduction of L8-BO-F as an additional electron transport component is desirable to improve the electron transport properties.

The exciton generation contour was simulated from FLAS in combination with optical matrix-transfer models (Fig. 3d–f). For the photovoltaic devices with around 120 nm-thick films, the excitons are mostly generated in the middle region of the active layer. After exciton dissociation, the hole and electron could respectively transport across tens of nanometers towards the electrodes. However, for the thick films, the excitons are mainly generated near PEDOT:PSS, since the light incidence direction is from PEDOT:PSS side (Supplementary Figs. 16 and 17). After exciton dissociation into holes and electrons, the electrons need to transport across a couples of hundreds of nanometers towards cathode, which indicates that the electron transport plays critical role in determining the final photovoltaic performance of thick-film devices. From the FLAS spectra, film-depth-dependent composition is extracted (Fig. 3g–i). For the 300 nm binary blend, the donor and acceptor have a good miscibility, showing composition profile with major BTP-eC9 at the top part (film-depth range 0–50 nm) of the film. For the ternary and LBL-processed ternary blends, the vertical phase distributions show slight fluctuation, as a result of dynamically varied solvent-gradient evolution during solvent evaporation[51,52]. In fact, the vertical phase separation could also be impacted by the confined space between the air-film and film-substrate interfaces, leading to a different phase in the infinity of the interfaces from that in the bulk. Particularly, for the LBL-processed ternary blend, graded vertical phase separation can be clearly identified, evidenced by the acceptor composition approaching 100% at the top surface (0 nm, active layer/PNDIT-F3N interface), which directly contacts the cathode. Schematic illustration of the vertical phase separation of different blends is illustrated in Fig. 3j–l. The acceptor-rich top surface can enhance the electron transport and electron charge collection at the cathode, which is favorable of the fabrication of thick-film OSCs.

Figure 4a–c shows the 2D GIWAXS pattern of the 300 nm-thick binary and ternary blends. Supplementary Fig. 18 shows the corresponding line-cut intensities taken along the in-plane (IP) and OOP direction. Both the binary and ternary blends adopt preferred face-on orientations, evidenced by the obvious lamellar stacking peak (100) in IP direction and the strong π–π stacking peak (010) in OOP direction. The corresponding CCLs associated with the lamellar and π–π stacking peaks in binary, ternary and LBL ternary blends are summarized in Supplementary Tables 5 and 6. We found that at 300 nm thickness, the CCLs of the ternary blend are larger than the host binary blend, manifesting

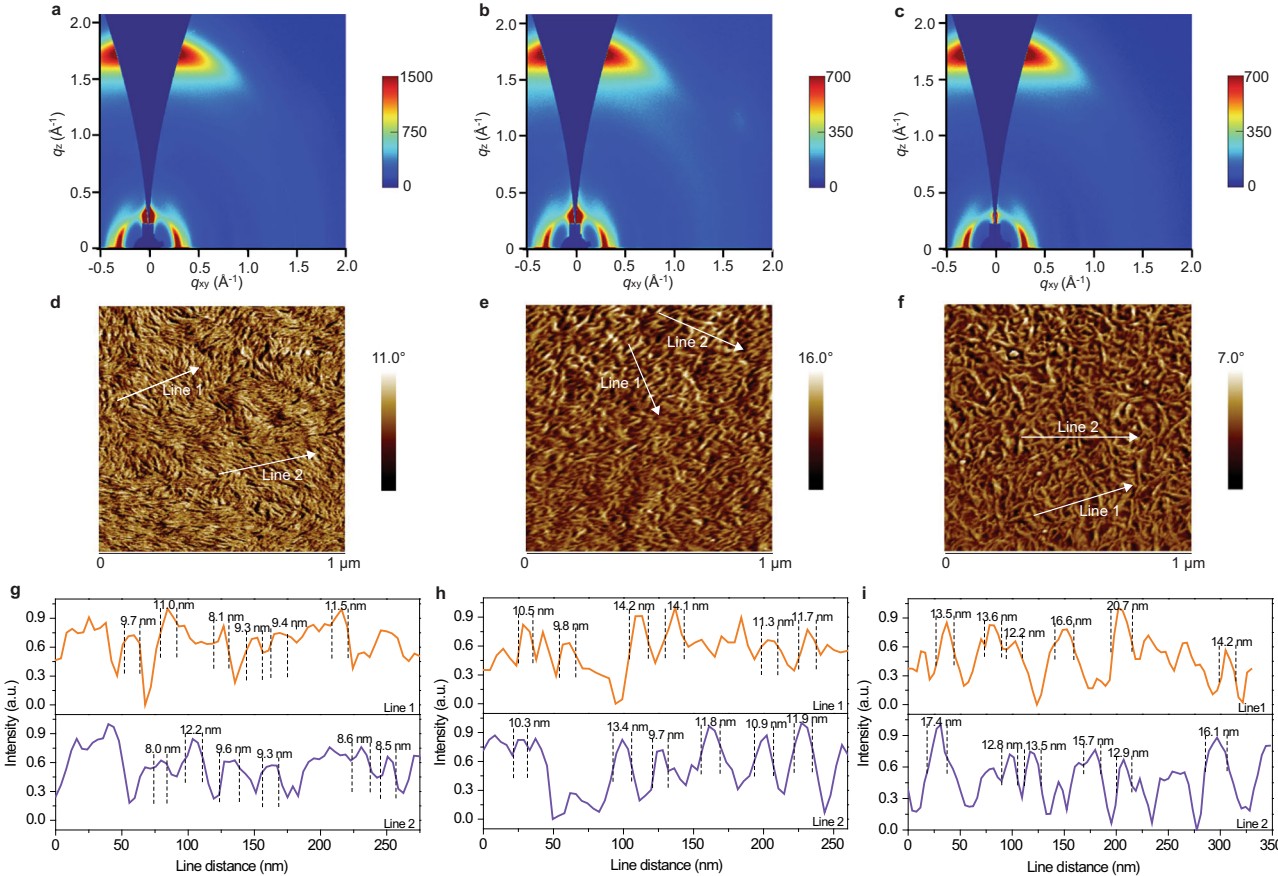

**Fig. 4 Morphology characterizations. a–c** 2D GIWAXS patterns of the (**a**) PM6:BTP-eC9 blend, **b** PM6:BTP-eC9:L8-BO-F blend, and (**c**) LBL-processed PM6:BTP-eC9:L8-BO-F blend with 300 nm thickness. **d–f** AFM phase images of (**d**) PM6:BTP-eC9 blend, **e** PM6:BTP-eC9:L8-BO-F blend, and (**f**) LBL-processed PM6:BTP-eC9:L8-BO-F blend with 300 nm thickness. **g–i** The line profiles to obtain the FWHM of cross-sections though AFM signals of (**g**) PM6:BTP-eC9 blend, **h** PM6:BTP-eC9:L8-BO-F blend, and (**i**) LBL-processed PM6:BTP-eC9:L8-BO-F blend (two lines are selected to calculate the FWHM).

that the addition of the third component leads to enhanced molecular crystallinity. In addition, more ordered molecular stacking and enhanced molecular crystallinity were achieved in LBL-processed ternary blend. Such findings agree well with the SCLC results as well. Further insights into the morphology of the binary and ternary blends were attained from atomic force microscopy (AFM) measurements. Figure 4d–f shows the phase images of the 300 nm-thick films and both the binary and ternary blends exhibit densely distributed fibrillar morphology, which is favorable of charge transport[53,54]. The height images of 300 nm-thick films are shown in Supplementary Fig. 19. The binary, ternary and LBL ternary blends exhibit root-mean-square values of 1.16, 1.27 and 1.73 nm, respectively. Notably, it was observed that the fibril width of the ternary blend is larger than the PM6:BTP-eC9 blend, and the fibril width further increases when using LBL method to prepare the film. Supplementary Figs. 20 and 21 present the AFM height and phase images of the binary, ternary and LBL ternary blends with 120 nm- and 500 nm-thick. Similar trends were also obtained. To make a detailed comparison, the fibril widths of the binary and ternary blends were measured. The line profiles of full-width half-maximum (FWHM) of the peaks in the AFM phase images are shown in Fig. 4g–i. The calculated average values are 9.6, 11.6 and 16.3 nm for the binary, ternary and LBL ternary blends, respectively, revealing that the different processing method indeed influences the molecular aggregation and the phase separation of the active layer.

**Charge recombination dynamics.** We also studied the charge recombination dynamics of the free polarons in the films by the ns-resolved TA spectroscopy. Figure 5 shows the TA data comparing the excited-state dynamics in the films with different thicknesses prepared with different methods. In the BHJ blends, the photogenerated excitons in the donor-acceptor films undergo charge transfer and charge separation processes to form free polarons within the nanosecond timescale. The major features of excited states are similar for both the PM6:BTP-eC9:L8-BO-F blend and LBL-processed PM6:BTP-eC9:L8-BO-F blend (Figs. 5a, b, 300 nm-thick film). The excited-state absorption (ESA) features in the ranges of 650–800 nm and >900 nm persisting to a longer timescale can be assigned to those of free polarons[55]. Considering nearly full absorption in the visible spectral range for the thick films, we take the dynamics curves probed in the infrared range at 940 nm as the benchmark to compare the dynamics in different samples (Fig. 5c). The decay half-life parameters recorded from the samples are summarized in Fig. 5d. The decay dynamics are largely the same for the 120 nm- and 300 nm-thick films prepared with both approaches, but exhibit significant difference for the 500 nm-thick film. With the thickness increasing from 120 nm to 300 nm, the carrier lifetime increases significantly in both the PM6:BTP-eC9:L8-BO-F and LBL-processed PM6:BTP-eC9:L8-BO-F films (Supplementary Fig. 22). When the thickness increases from 300 nm to 500 nm, no significant change is observed for the carrier lifetime of the LBL-processed film, but the carrier lifetime in the PM6:BTP-eC9:L8-BO-F film is

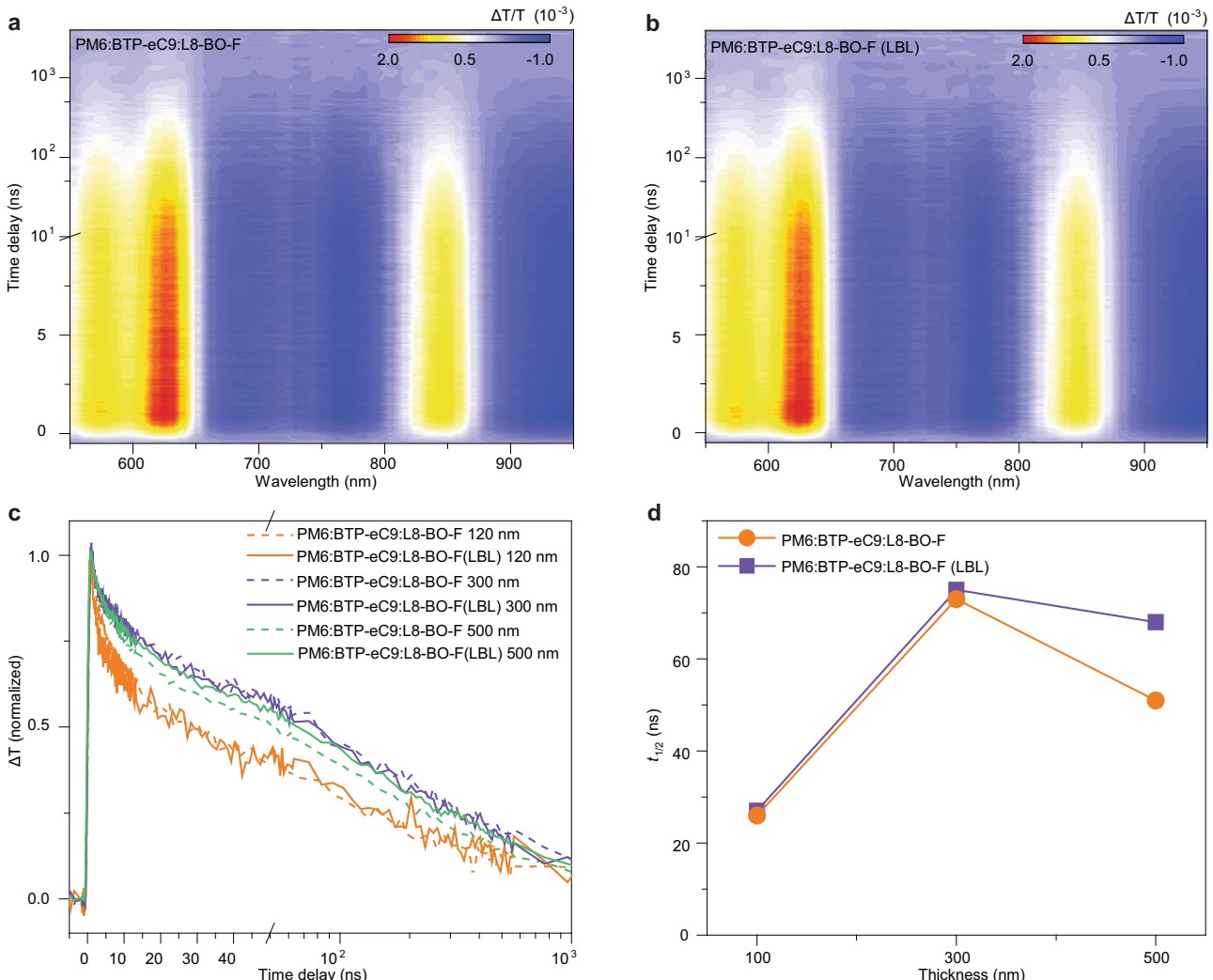

**Fig. 5 Charge recombination dynamics.** TA data recorded from (**a**) 300 nm-thick PM6:BTP-eC9:L8-BO-F blend. **b** 300 nm-thick LBL-processed PM6:BTP-eC9:L8-BO-F blend. **c** Normalized time-resolved curves probed at 940 nm in the films of different thicknesses prepared by different methods. **d** The half-decay lifetimes estimated from the ns kinetic curves for different samples.

markedly reduced. The increased trap-assisted recombination in the 500 nm-thick devices may account for the reduction of carrier lifetime[56,57]. These results demonstrate that the bulk recombination of electron and hole polarons decreases, benefiting from the layered device structure, which is plausibly another reason for the better efficiency retaining in the LBL-processed devices with thick active layers.

**Energy loss analysis.** Finally, we evaluated the voltage loss ($V_{loss}$) in the 300 nm-thick binary and ternary devices, using the theory derived in the framework of Marcus theory[58]. The detailed extracted values are summarized in Supplementary Table 7. Supplementary Fig. 23 displays the electroluminescence (EL) spectra of the blend films. The optical bandgaps ($E_g$s) of the active layers were determined by finding the crossing point between the normalized reduced PL and absorption spectra[59]. We found that the $E_g$s of the binary, ternary and LBL-processed ternary blends were all 1.42 eV (Supplementary Fig. 24). The total voltage losses ($V_{loss}$s) for the binary, ternary and LBL-processed ternary systems were calculated to be 0.60, 0.58 and 0.58V, respectively. In general, $V_{loss}$ of a solar cell consists of radiative recombination ($\Delta V_{rad}$) and non-radiative recombination ($\Delta V_{non-rad}$). $\Delta V_{rad}$ is the voltage loss caused by radiative recombination. $\Delta V_{non-rad}$ is the voltage loss

that originates from non-radiative recombination, which is a key factor in determining the $V_{oc}$ of the OSCs. $\Delta V_{non-rad}$ can be calculated from the equation of $\Delta V_{non-rad} = \frac{kT}{q}\ln\left(\frac{1}{EQE_{EL}}\right)$, where $EQE_{EL}$ is electroluminescence external quantum efficiency[60]. As plotted in Supplementary Fig. 25, the $EQE_{EL}$ values of the binary, ternary and LBL-processed ternary OSCs are $9.5 \times 10^{-5}$, $1.1 \times 10^{-4}$ and $2.0 \times 10^{-4}$, respectively, and the corresponding $\Delta V_{non-rad}$s were calculated to be 0.23, 0.23, and 0.21 V. Then the $\Delta V_{rad}$ of the solar cells could be determined, which are 0.37, 0.35 and 0.37 V for the binary, ternary and LBL-processed ternary blends, respectively. From these results, we can conclude that the introduction of the third component can afford reduced $V_{loss}$, and therefore increase the $V_{oc}$ of the thick devices. Additionally, LBL processing method could enable the OSCs with relatively lower $\Delta V_{non-rad}$.

The aforementioned results highlight the role of exciton diffusion length in thick-film OSCs based on two NFAs. To enhance the exciton diffusion, the following material selection criteria should be taken into account: (1) efficient energy transfer should occur between the two NFAs, which can facilitate exciton hopping among conjugated segments in organic materials; (2) the degree of crystallinity of the host NFA should be increased after the addition of another NFA, which can lower the energetic

disorder in materials and thereby favor the exciton diffusion. Following this selection guideline, we selected Y6 and Y6-F acceptors to fabricate thick-film OSCs. As shown in Supplementary Figs. 26, 27 and Table 8, efficient Förster energy transfer from Y6-F to Y6 occurs and the crystalline order is much improved when Y6-F blended with Y6. As a result, the calculated $L_D$ in the blend film is 37.9 nm, higher than 30.7 nm in neat Y6 film, and 37.4 nm in neat Y6-F film (Supplementary Figs. 28 and Table 9). We then fabricated thick-film OSCs based on Y6 and Y6-F. As expected, compared to the binary OSCs, PM6:Y6:Y6-F ternary device demonstrates 4.17%, 6.10% and 6.23% improvements in efficiency at the active layer thickness of 120 nm, 300 nm, and 500 nm, respectively (Supplementary Figs. 29, 30 and Table 10). Additionally, the LBL strategy can further improve the efficiency of thick-film OSCs and 300 nm- and 500 nm-thick PM6:Y6:Y6-F devices display higher efficiencies of 15.42% and 14.70%, respectively (Supplementary Fig. 31).

## Discussion

In conclusion, we have been able to fabricate a series of efficient thick-film OSCs based on two NFAs. Among them, the 300 nm-thick device based on PM6:BTP-eC9:L8-BO-F blend shows a PCE of 17.31%. Even increasing the film thickness to 500 nm, the device can still obtain a PCE of 15.21%. Such encouraging achievements are resulted from the vertically optimized phase separation via LBL processing that can ensure highly effective charge transport and charge collection by the electrodes, and enlarged exciton diffusion length in the mixed acceptor domain that can favor long-range exciton diffusion. As a result, improved exciton dissociation, enhanced charge transport, and suppressed trap-assisted recombination contribute to the high $J_{sc}$, $V_{oc}$, and FF achieved in the thick-film OSCs. Our work affords a useful material selection criteria to enhance exciton diffusion and a general device processing approach for the fabrication of thick-film OSCs, which takes a further step forward the practical applications of OSCs.

## Methods

**Materials**. PM6 was purchased from Solarmer Materials Inc. BTP-eC9 and Y6 were purchased from hyperchemical Inc. L8-BO-F and Y6-F were synthesized in our group. Chloroform, chloronaphthalene and 1,8-Diiodooctane were purchased from Sigma Aldrich Inc.

**Optolelectronic characterization**. UV–vis absorption spectra of the pristine and blend films were measured with a UV–vis spectrophotometer (Shimadzu UV-3700).

**Device fabrication and measurement**. Prepatterned ITO-coated glass substrates were cleaned with detergent and ultrasonicated in deionized water, acetone and isopropanol for 15 min respectively and then dried in an oven overnight. Before use, ITO-coated substrates were treated by plasma for 2 min PEDOT:PSS (Heraeus Clevios P VP. AI 4083, filtered at 0.45 µm) was spin-coated onto the ITO surface at 4000 rpm for 30 s, and thermal annealed at 150 °C for 10 min in air. The binary and ternary devices were fabricated with a conventional device structure of ITO/PEDOT:PSS/ active layer/PNDIT-F3N/Ag. For all the devices, the active layers were formed by spin coating a chloroform solution of active layer in a N$_2$-filled glove box. The total D/A weight ratio was kept at 1:1.2 in PM6:BTP-eC9, PM6:BTP-eC9:L8-BO-F, PM6:Y6 and PM6:Y6:Y6-F blends. For PM6:BTP-eC9 and PM6:BTP-eC9:L8-BO-F blends, the PM6 concentrations are 7 mg/ml, 12.8 mg/ml and 15.3 mg/ml for 120 nm-, 300 nm- and 500 nm-thick devices, respectively. 0.5% DIO was used as the solvent additive for these mixed solutions. For PM6:Y6 and PM6:Y6:Y6-F blends, the PM6 concentrations are 7 mg/ml, 12.8 mg/ml, and 15.3 mg/ml for 120 nm-, 300 nm- and 500 nm-thick devices, respectively, and 0.5% CN was used as the solvent additive. For the LBL-processed ternary devices, the PM6 dissolved in chloroform were firstly spin cast onto the PEDOT:PSS substrates with the same concentrations mentioned above for the fabrication of the conventional devices. Then the acceptors in chloroform with 0.5% DIO or CN solvent additives (DIO for BTP-eC9:L8-BO-F, and CN for Y6:Y6-F) were spin coated on the top of the donor layers to form active layers. The concentrations of BTP-eC9 and BTP-eC9:L8-BO-F for 120 nm-, 300 nm-, 500 nm-thick devices are 8.4 mg/ml, 15.4 mg/ml, and 18.4 mg/ml, respectively. The concentrations of Y6 and Y6:Y6-F are the same with BTP-ec9 and BTP-eC9:L8-BO-F. All of the active layers

were thermal annealed at 100 °C for 10 min. Subsequently, the methanol solution of PNDIT-F3N (0.6 mg/mL) was spin coated on the top of active layer (3500 rpm). At last, 100 nm-thick Ag was thermally deposited under the vacuum condition of $3 \times 10^{-4}$ Pa. The active area of the devices is 4.0 mm$^2$, and the mask area is 3.152 mm$^2$. For the large-area devices, the active area is 1.0 cm$^2$, and the mask area is 0.982 cm$^2$. The $J$–$V$ curves were measured from −0.5 to 1 V with a scan step of 50 mV and a dwell time of 10 ms, along the forward scan direction, using a Keithley 2400 Source Measure Unit. The photovoltaic performance of all the OSCs was measured in a N$_2$-filled glove box at room temperature (ca. 25 Celsius degree) using an Air Mass 1.5 Global (AM 1.5 G) solar simulator (SS-F5-3A, Enlitech) with an irradiation intensity of 100 mWcm$^{-2}$, which was measured by a calibrated silicon solar cell (SRC2020, Enlitech). The EQE spectra were measured from a QEX10 Solar Cell EQE measurement system (QE-R3011, Enlitech).

**Photostability test**. Photostability of the devices were tested under Maximum Power Point Tracking with 1-sun illumination (white LED lamp). At the beginning of the test, the bias voltage is usually set at 0 V and the disturbance step is 0.01 V. As the test goes on, the bias voltage setting and step can be changed automatically, until the bias voltage approaches to the maximum power point voltage $V_{max}$.

**Carrier mobility measurement**. Carrier mobility was measured by the SCLC method. The mobility was determined by fitting the dark current to the model of a single carrier SCLC, according to the equation of $J = 9\varepsilon_0\varepsilon_r\mu V^2/8d^3$, where $J$ is the current density, $\varepsilon_0$ is the permittivity of free space, $\varepsilon_r$ is the relative dielectric constant of the transport medium, $\mu$ is the charge carrier mobility and d is the film thickness of the active layer. The carrier mobility can be calculated from the slope of the $J^{0.5}$~$V$ curves.

**Morphology characterization**. GIWAXS measurements were carried out at the PLS-II 9A U-SAXS beamline of the Pohang Accelerator Laboratory in Korea. AFM measurements were performed on a Dimension Icon AFM (Bruker) in a tapping mode under ambient conditions.

**Film-depth-dependent light absorption spectra**. Film-depth-dependent light absorption spectroscopy was measured by a homemade setup. Soft plasma generated by low-pressure (less than 20 Pa) oxygen was used for the incremental etching of the organic film. The UV–vis absorption spectrum during each etching was in-situ monitored by an optical spectrometer. Beer–Lambert's law was utilized to fit the film-depth-dependent light absorption spectra, which were subsequently utilized to fit the exciton generation contour upon a modified optical matrix-transfer approach[48].

**Transient absorption spectroscopy**. The femtosecond TA measurements were conducted using a Yb:KGW laser (Pharos, Light Conversion). The wavelength of fundamental output was at ~1030 nm. We used a homebuilt noncollinear optical parametric amplifier to generate the pump pulses centered at 800 nm. The probe beam was supercontinuum generated by focusing a small fraction of the fundamental 1,030 beam to a 5 mm sapphire plate. The supercontinuum light was split into two beams for balanced detection. using a double-line Si camera (S14417, Hamamatsu) mounted on a monochromator (Acton 2358, Princeton Instrument). Pulse-to-pulse spectral analysis was conducted at the rate of 50 kHz enabled by a homebuilt field-programmable gate array (FPGA) control board. The noise level (ΔT/T) was better than $10^{-6}$ after averaging 25K pump-on and pump-off shots for each data point. For ns-resolved TA spectroscopy, the pump laser was replaced by a pulsed laser diode emitted at 670 nm (LDH-P-C-670M, Picoquant). The time delay between the two lasers was synchronized and enabled by a digital delay generator (DG645, Stanford Research System). The samples were kept in nitrogen atmosphere during the measurement to minimize photon-induced degradation.

**Photoluminescence spectra and time-resolved photoluminescence**. Photoluminescence (PL) spectra and time-resolved photoluminescence (TRPL) dynamics were acquired using a homemade confocal microscopy-based spectroscopy system. An 800 nm pulse (80 MHz, 35 fs) was generated by a Ti:sapphire amplifier (Maitai HP, Spectra Physics). Then the 800 nm pulse was guided into a frequency doubler to generate 400 nm pulse, which was further coupled into a confocal microscopy system (Nanofinder FLEX2, Tokyo Instruments, Inc.) to excite the film samples. The PL spectra were recorded by a fiber-connected spectrometer (DU420A-OE, ANDOR) to record the PL spectra, and the TRPL kinetics were collected by a TCSPC module (Becker & Hickl, SPC-150). The pump power density was fixed at 5 µJ cm$^{-2}$. All samples were fabricated on quartz substrates and were encapsulated in a nitrogen glove-box with epoxy resin.

**Voltage loss characterization**. The electroluminescence spectra were obtained using a Kymera-328I spectrograph and a Si EMCCD purchased from Andor Technology (DU970P), and a InGaAs CCD camera (DU491A-1.7, Andor Technology). Injection current used for EL was 1 mA. To determine EQE$_{EL}$, we first performed electroluminescence measurements, and found that the emission peaks are at 1.3–1.4 eV (Supplementary Fig. 22), for all of the active layers studied in this

work. Then, we determine EQE$_{EL}$ using a homebuilt setup using a Keithley 2400 to inject current to the solar cells. Emission photon-flux from the solar cells was recorded using a Si detector (Hamamatsu s1337-1010BQ) and a Keithley 6482 picoammeter.

## Data availability

Source data are provided with this paper. All other data generated or analyzed during this study are included in the published article and its Supplementary Information.

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

## Acknowledgements

This work was financially supported by the National Natural Science Foundation of China (NSFC) (Grant Nos. 51825301, 52003013, 21734001, and 51873172). G.H.L. acknowledges the financial support from the Shanxi Provincial Key R&D Program (Grant No. 2021GXLH-Z-055). Z.T. acknowledges the financial support from the Shanghai Pujiang Program (Grant No. 19PJ1400500). H.Y.W acknowledges the financial support by the National Research Foundation (NRF) of Korea (2019R1A2C2085290, and 2020M3H4A3081814).

## Author contributions

Y.C. fabricated and characterized the devices. Q.L. and C.Z. measured TAS and performed the analysis. G.Y.L. and G.H.L. measured the film-depth-dependent light absorption spectra and performed the optical simulation. H.S.R. and H.Y.W. performed the GIWAXS measurements. Y.L. synthesized L8-BO-F and Y6-F. H.J. and Z.T. studied the energy loss of the devices. Z.C. and X.H. performed photoluminescence measurements. G.H.L., C.Z. and Y.S. supervised and directed this project. Y.C. and Y.S. wrote the manuscript. All authors commented on the manuscript.

## Competing interests

The authors declare no competing interests.
