## [Peer Review File · Nature Communications]

Reviewer comments, initial review –

Reviewer #1 (Remarks to the Author):

This manuscript reports organic solar cells with thick active layers with efficiencies exceeding 17% with thickness in excess of 300 nm. The results obtained, especially the layer-by-layer-based OPV, are impressive and will attract interest in the OPV community. All measurements are coherent with each other and well-performed, spanning from optoelectronic characterisations as well as morphological investigations. Thick active layers are of the utmost importance for scaling up OPV technology, and this work set the basis for it. I recommend publishing the paper in Nature Communications after the revisions suggested.

- 1) The FRET mechanism needs to be clarified. The overlapping between absorption and emission reported in Supplementary Figure 12, do not show a sufficient overlap to exclude charge transfer. Moreover, the steady-state PL can give only a qualitative idea about the mechanism. I suggest performing transient PL to conclude the mechanism behind this ternary blend.
- 2) Towards up-scaling production, i.e. R2R, thin interlayers cannot be used. It would be great to see these devices in a R2R-compatible device stack consisting of all-solution processed layers except the top electrode.
- 3) How is the photostability of these thick devices compared to thin devices?
- 4) In TA results, very thick layers have been used. Can the authors comment on that, especially considering avoiding the probe being all absorbed by the photoactive layer?
- 5) Worth mentioning works conducted on fullerene-based thick OPV.

Reviewer #2 (Remarks to the Author):

In this manuscript, Cai et al. developed a series of BHJ or LBL ternary organic solar cells (OSCs) and reached a high efficiency of 17.31% for 300 nm thick OSC and 15.21% for 500 nm OSC. However, the obtained efficiency for 300 nm thick LBL ternary OSC is marginally improved than that of the BHJ ternary one (16.92%). Meanwhile, although lots of works have been done, my concern mainly lies in the novelty of the research. Using a ternary strategy for thick-film OSC has been successfully proved by [Nat. Energy 2018, 3, 952-959]. Especially, the same materials and combination, PM6:BTP-eC9:L8-BO-F, have been repeatedly used in other works by the same authors [Adv. Mater. 2021, 33, 2101733] [Energy Environ. Sci., 2021,14, 5009-5016]. In the reported work {Adv. Mater. 2021, 33, 2101733}, the same research group has systematically studied the PM6:BTP-eC9:L8-BO-F-based ternary system and obtained high PCEs. The lack of novelty makes this manuscript not suitable for publishing in Nat. Commun. Some important problems are not explained clearly. Detailed comments are provided below:

1. The Jscs declined dramatically when the thickness increased from 300 nm to 500 nm, but the authors failed to give an explanation. The reduced carrier lifetime might partly explain these results but seems not fully interpreted by the authors, and the reasons that caused the reduction of carrier lifetime were not studied in this manuscript, too. The authors are recommended to add some relevant studies.
2. In Fig. 3g-i, the authors gave the film-depth-depended composition of different blends, they claimed the binary blend "showing composition profile with major BTP-eC9 at the top part (film depth range 0-150 nm) of the film". It is true for the top dozens of nanometers, but after 50 nm, the ratio of donor and acceptor is near the ratio in PM6:BTP-eC9 solution (1:1.2), so the depth range

where BTP-eC9 is dominant should be reconsidered.

3. It was noticed that, according to Fig 3, the D/A ratio for all blends did not change linearly, especially for ternary blends at the depth range of 150-230 nm; what's the reason that caused this result from the view of dynamics? The authors need to provide other evidence to support these results.

4. The energy loss analysis in this manuscript is apparently incomplete, and figures that determined optical bandgaps were not presented.

5. In general, when the emission device becomes saturation, EQE_EL should have a constant value. The authors' EQE_EL changes with the current. The authors seem to have calculated non-radiative losses using the values obtained at 1E-4 A. If calculated from 0.01 A, the EQE_EL values of the ternary and LBL-processed ternary OSCs are very close. The detailed values of the other energy loss terms will be changed. What is the basis for using the 1E-4 A for EQE_EL calculation?

6. In Supplementary Figure 21, there are 3 or 4 obvious peaks in the normalized EL curves, which have been rarely seen in other papers; what are the causes for this phenomenon?

7. Compared with BTP-eC9, L8 with a branched side-chain may possess less crystallinity, and why the introduction of L8 into the BTP-eC9-based active layer would improve the crystallinity of the acceptor phase. The lack of reasonable explanations discredited the results.

8. As shown in the absorption spectra (Figure S1), when focusing on the absorption in 500-700 nm for the polymer donor, that in the LBL film is much lower than that in the BHJ one. It's hard to believe that the LBL and BHJ ternary devices delivered almost overlapped EQE responses (Fig. 1d).

9. In this article, there exist lots of deficiencies in language and grammar; for example, in Line 112-114, Page 5, the sentence "Our work not only highlights the role of....." seems incomplete; the indefinite article before FF should be "an" (Line 129-130, Page 6).

Reviewer #3 (Remarks to the Author):

In the ms, the authors reported the fabrication of thick-film OSCs using the ternary device strategy. A PCE of 17.31% was achieved for the 300 nm-thick device based on PM6:BTP-eC9:L8-BO-F blend. With 500 nm active layer, the device can deliver a PCE of 15.21%. The good devices results were thought to be ascribe to the increased exciton length for the ternary devices. In addition, the authors also gave some criteria for the active layer selection to fabrication thick film devices. Thick film devices with high efficiencies are indeed favorable for the large scale production of OSCs. This work can give some valuable insights for fabrication thick film OSCs active layers. It can be accepted after considering the following detailed comments.

1.As emphasized in the ms, the exciton diffusion length played the critical role for thick film device. What is the factors that determine the exciton diffusion length.

2.In the paragraph started from line 382, the authors gave some suggestions on the acceptor selection to obtain the enhanced exciton diffusion length. The role of the polymer donor is not mentioned. It should also play an important role on the exciton diffusion length. More comments and explanation on it should be given.

3.Large area device at least 1cm² are suggested to give.

4.The device stability measurement, at least some initial results are recommended to give for the thick film device.

Responses to the reports of the Reviewers

Reviewers' Comments:

Reviewer #1 (Remarks to the Author):

This manuscript reports organic solar cells with thick active layers with efficiencies exceeding 17% with thickness in excess of 300 nm. The results obtained, especially the layer-by-layer-based OPV, are impressive and will attract interest in the OPV community. All measurements are coherent with each other and well-performed, spanning from optoelectronic characterizations as well as morphological investigations. Thick active layers are of the utmost importance for scaling up OPV technology, and this work set the basis for it. I recommend publishing the paper in Nature Communications after the revisions suggested.

Comment 1: The FRET mechanism needs to be clarified. The overlapping between absorption and emission reported in Supplementary Figure 12, do not show a sufficient overlap to exclude charge transfer. Moreover, the steady-state PL can give only a qualitative idea about the mechanism. I suggest performing transient PL to conclude the mechanism behind this ternary blend.

Reply 1: Thanks for the comments. To study the charge transfer behavior, OSCs based on BTP-eC9, L8-BO-F and BTP-eC9:L8-BO-F films were fabricated. As shown in Figure R1a, the BTP-eC9:L8-BO-F device yields a J_{sc} of 0.11 mA cm^{-2} , which is smaller than that (0.28 mA cm^{-2}) of the OSC based on neat BTP-eC9, indicating negligible charge transfer between BTP-eC9 and L8-BO-F. Moreover, time-resolved photoluminescence (TRPL) experiments were performed to further study the charge/energy transfer between BTP-eC9 and L8-BO-F. After adding the L8-BO-F to the host BTP-eC9 acceptor, the average fluorescence lifetime (τ) is 824 ps at 900 nm, much longer than that (554 ps at 900 nm) of the neat BTP-eC9 film (Figure R1b). Taken together, these results suggest that the fast Förster energy transfer of excitons from L8-BO-F to BTP-eC9 occurs in their blend. We added Figure R1 to Supplementary Figure 14 and the corresponding descriptions were added to the main text in the revised manuscript.

Figure R1. (a) J - V characteristics of OSCs based on BTP-eC9, L8-BO-F and BTP-eC9:L8-BO-F films. (b) TRPL spectra of BTP-eC9 and BTP-eC9:L8-BO-F films.

Comment 2: Towards up-scaling production, i.e. R2R, thin interlayers cannot be used. It would be great to see these devices in a R2R-compatible device stack consisting of all-solution processed layers except the top electrode.

Reply 2: Thanks for the suggestions. As we mentioned in our manuscript, the development of high-performance OSCs with thick active layers is of crucial importance for the roll-to-roll printing of large-area solar panels. Here, we fabricated devices in R2R-compatible device stack consisting of all-solution processed layers except the top electrode. As shown in Figure R2, a 30-nm PEI-Zn as the ETL layer was deposited on the ITO glass by blade coating with a coating speed of 10 mm/s and a blade-substrate gap of 150 μm in air, followed by thermal annealing at 150 $^{\circ}\text{C}$ for 10 min. Then, the active layer was blade-coated with a coating speed of 5 mm/s and a blade-substrate gap of 150 μm in air. The prepared samples were transferred to a N_2 -filled glove box and annealed at 100 $^{\circ}\text{C}$ for 10 min. A 30-nm PEDOT as the HTL layer was then blade-coated with a coating speed of 10 mm/s and a blade-substrate gap of 100 μm in air. Finally, Ag (70 nm) was evaporated through a shadow mask. The effective device area is 0.04 cm^2 . The fabricated OSCs showed PCEs of 16.26% with a 120-nm active layer film and 15.36% with a 300-nm active layer film (Table R1). These results indicate that the thick-film ternary strategy has great potential in industrial R2R production of OSCs

Figure R2. Schematic representations of the doctor-blade method, the device structure and the J - V characteristics of the corresponding OSCs.

Table R1. Summary of device parameters of the doctor-blade coated OSCs with different active layer thicknesses.

Active layer	V_{oc} (V)	J_{sc} (mA cm^{-2})	FF	PCE (%)
PM6:BTP-eC9:L8-BO-F (~120 nm)	0.84	26.17	0.74	16.26
PM6:BTP-eC9:L8-BO-F (~300 nm)	0.83	27.22	0.68	15.36

Comment 3: How is the photostability of these thick devices compared to thin devices?

Reply 3: Photostability of the thin and thick devices was measured under continuous 1 sun illumination in air. As shown in Figure R3, 120 nm-, 300 nm- and LBL-processed 300 nm-thick ternary devices reached their T_{80} (the time required to reach 80% of initial performance) at 127, 138, and 152 h, respectively. The results indicate that the thick devices exhibit slightly

improved photostability than the thin devices. On the other hand, more ordered molecular stacking, enhanced crystallinity, and reduced trap state density in LBL-processed thick film compared to the conventional thick film help increase the photostability of OSCs. The results and the discussions regarding the photostability of thick and thin devices were also added in the revised manuscript.

Figure R3. Photostability of 120 nm- and 300 nm-thick ternary devices measured under continuous 1 sun illumination in air.

Comment 4: In TA results, very thick layers have been used. Can the authors comment on that, especially considering avoiding the probe being all absorbed by the photoactive layer?

Reply 4: Indeed, most probe light is absorbed by thick active layers. In the strong absorbing wavelength range, only ~ 1% of the probe light can penetrate through the sample. In order to detect the transient absorption (TA) signal with weak probe, we use a home-built highly sensitive spectral analyzer using a silicon image sensor (S14417, Hamamatsu). Being enabled by a FPGA-based circuit, the analyzer can work at 50 kHz, which is much faster than the sensors used in commercially available TA systems. The TA data were captured using the pulse-to-pulse spectral analysis with 25 k pump-on and pump-off shots for each data point. The measurements were repeated for 20 times to obtain a relatively good signal-to-noise ratio.

Comment 5: Worth mentioning works conducted on fullerene-based thick OPV.

Reply 5: The imbalanced charge transport due to the high electron mobility of fullerene acceptor and the relatively low hole mobility of the polymer donor, limits the further efficiency improvements for the fullerene-based thick OSCs. The highest efficiency of fullerene-based OSCs is 11.3% with the active layer thickness of 280 nm (SusMat. 2021, 1, 4–23; J. Mater. Chem. A 2021, 9, 3125-3150). The descriptions about the fullerene-based

thick OSCs were also added to the main text in the revised manuscript.

Reviewer #2 (Remarks to the Author):

In this manuscript, Cai et al. developed a series of BHJ or LBL ternary organic solar cells (OSCs) and reached a high efficiency of 17.31% for 300 nm thick OSC and 15.21% for 500 nm OSC. However, the obtained efficiency for 300 nm thick LBL ternary OSC is marginally improved than that of the BHJ ternary one (16.92%).

Comment 1: Meanwhile, although lots of works have been done, my concern mainly lies in the novelty of the research. Using a ternary strategy for thick-film OSC has been successfully proved by [Nat. Energy 2018, 3, 952-959]. Especially, the same materials and combination, PM6:BTP-eC9:L8-BO-F, have been repeatedly used in other works by the same authors [Adv. Mater. 2021, 33, 2101733] [Energy Environ. Sci., 2021,14, 5009-5016]. In the reported work {Adv. Mater. 2021, 33, 2101733}, the same research group has systematically studied the PM6:BTP-eC9:L8-BO-F-based ternary system and obtained high PCEs. The lack of novelty makes this manuscript not suitable for publishing in Nat. Commun.

Reply 1: We greatly thank the reviewer for her/his effort in evaluating our manuscript and for providing valuable comments. However, we politely disagree with the conclusion made by the reviewer that this work is lack of novelty. We would like to take this opportunity to better explain the novelty and significance of our work as follows:

1. The development of high-performance OSCs with thick active layers is crucially important for the roll-to-roll printing of large-area solar panels. Unfortunately, increasing the active layer thickness usually results in a significant reduction in efficiency. Here we report record high efficiencies of 17.31% (certified value of 16.9%) and 15.21% for the 300 nm- and 500 nm-thick OSCs made of one donor and two NFAs, respectively.

2. The two NFAs were interestingly found to possess enlarged exciton diffusion length up to 47 nm in the mixed phase compared to their neat films. Such long-range exciton transport is favorable of charge carrier generation and extraction in the thick-film OSCs. We reveal the underlying reason for the improved exciton diffusion length and provide a general material selection guideline: (1) efficient energy transfer should occur between the two NFAs, which can facilitate exciton hopping among conjugated segments in organic materials; (2) the degree of crystallinity of the host NFA should be increased after the addition of another NFA, which can lower the energetic disorder in materials and thereby a longer exciton diffusion is expected. We further use this selection guideline to select another ternary system and successfully fabricate high-efficiency thick-film OSCs.

3. Several photovoltaic material systems show that the layer by layer processed devices

show much higher efficiencies than the conventional counterparts, indicating that the general application of layer by layer approach in further improving the efficiency of thick-film OSCs by optimizing the vertical phase separation of the active layers.

Based on the above-mentioned results, it can be seen clearly that our work differs significantly with the work reported in Nat. Energy 2018, 3, 952-959, in which the authors mainly highlight the importance of a hierarchical morphology consisting of a PCBM transporting highway and an intricate non-fullerene phase-separated pathway network in all-small-molecule ternary blend, as well as the work reported in Adv. Mater. 2021, 33, 2101733 and Energy Environ. Sci. 2021, 14, 5009-5016, in which thin ternary blends have been employed to obtain high efficiencies.

Just as the quite positive evaluations of our manuscript from Referee #1: "Thick active layers are of the utmost importance for scaling up OPV technology, and this work set the basis for it", and Referee #3: "Thick film devices with high efficiencies are indeed favorable for the large scale production of OSCs. This work can give some valuable insights for fabrication thick film OSCs active layers", our work provides a useful material selection criteria to enhance exciton diffusion and a general device processing approach for the fabrication of high-efficiency thick-film OSCs, which is of great importance and interests to the OPV community.

Comment 2: The J_{sc} s declined dramatically when the thickness increased from 300 nm to 500 nm, but the authors failed to give an explanation. The reduced carrier lifetime might partly explain these results but seems not fully interpreted by the authors, and the reasons that caused the reduction of carrier lifetime were not studied in this manuscript, too. The authors are recommended to add some relevant studies.

Reply 2: We agree with the reviewer that the reduced carrier lifetime is probably related to the decline of short-circuit current. Indeed, the carrier lifetime (τ) is particularly important for devices with thick active layers. The carrier diffusion length (L) is dependent on the carrier mobility (μ) and lifetime, i.e., $L \sim (\mu\tau)^{1/2}$. The reduced carrier lifetime may significantly reduce the fraction of carriers that can be captured by electrodes, resulting in decreasing the short-circuit current. The carrier lifetime differences in different samples are on the time scale of tens of ns or longer, which is likely to be related to the trap states in the samples. The results of dependence of V_{oc} on the light intensity (P_{light}) show that the slope values are 1.37, 1.32 and 1.27 kT/q for the 300 nm-thick binary, ternary and LBL-processed ternary devices, respectively, while the slope values are 1.53, 1.45 and 1.38 kT/q for the 500

nm-thick OSCs correspondingly (Supplementary Fig. 12). The results indicate the increased trap-assisted recombination in the 500 nm-thick devices, which leads to the reduction of carrier lifetime (Adv. Mater. 2021, 2101833; J. Mater. Chem. A, 2021, 9, 13515-1352). The related discussions for the reasons that caused the reduction of carrier lifetime were also added in the revised manuscript.

Comment 3: In Fig. 3g-i, the authors gave the film-depth-depended composition of different blends, they claimed the binary blend "showing composition profile with major BTP-eC9 at the top part (film depth range 0-150 nm) of the film". It is true for the top dozens of nanometers, but after 50 nm, the ratio of donor and acceptor is near the ratio in PM6:BTP-eC9 solution (1:1.2), so the depth range where BTP-eC9 is dominant should be reconsidered.

Reply 3: We agree with the reviewer that the description of "showing composition profile with major BTP-eC9 at the top part (film depth range 0-150 nm) of the film" is not very accurate. We changed "film depth range 0-150 nm" to "film depth range 0-50 nm" in the revised manuscript.

Comment 4: It was noticed that, according to Fig 3, the D/A ratio for all blends did not change linearly, especially for ternary blends at the depth range of 150-230 nm; what's the reason that caused this result from the view of dynamics? The authors need to provide other evidence to support these results.

Reply 4: Thanks for the comments. In fact, like in-plane lateral phase separation of binary or ternary blends, vertical phase separation with separation features varying from nanometers or micrometers is also a commonly observed phenomenon in many immiscible polymeric blends. The nonlinear distribution of thick donor/acceptor active layer along the film-depth direction has been frequently observed in some other solution-proceeded heterojunctions (ACS Nano, 2011, 5, 1, 329-336; J. Mater. Chem. A 2016, 4, 15522 – 15535), as a result of dynamically varied solvent-gradient evolution during solvent evaporation. In fact, the vertical phase separation could also be impacted by the confined space between the air-film and film-substrate interfaces, leading to a different phase in the infinity of the interfaces from that in the bulk. The related discussions for the reasons that caused this result from the view of dynamics were also added in the revised manuscript.

Comment 5: The energy loss analysis in this manuscript is apparently incomplete, and figures

that determined optical bandgaps were not presented.

Reply 5: Thanks for the comments. The optical bandgaps of the active layers were determined by finding the crossing point between the normalized reduced PL and absorption spectra (Sustainable Energy Fuels, 2018,2, 538-544), which has been shown in Figure R4. Moreover, the detailed energy loss of OSCs based on PM6:BTP-eC9 and PM6:BTP-eC9:L8-BO-F blends is summarized in Table R2 and the corresponding descriptions including Figure R4 and Table R2 were added in the revised manuscript.

Figure R4. Normalized reduced absorption and PL spectra of (a) PM6:BTP-eC9, (b) PM6:BTP-eC9:L8-BO-F, and (c) LBL-processed PM6:BTP-eC9:L8-BO-F films.

Table R2. Detailed energy loss of OSCs based on PM6:BTP-eC9 and PM6:BTP-eC9:L8-BO-F blends.

Active layer	V_{oc} [V]	E_g^a [eV]	V_{loss} [V]	ΔV_{rad}^b [V]	$\Delta V_{non-rad}^c$ [V]	EQE _{EL}
PM6:BTP-eC9:L8-BO-F	0.82	1.42	0.60	0.37	0.23	9.5×10^{-5}
PM6:BTP-eC9:L8-BO-F	0.84	1.42	0.58	0.35	0.23	1.1×10^{-4}
PM6:BTP-eC9:L8-BO-F (LBL)	0.84	1.42	0.58	0.37	0.21	2.0×10^{-4}

^a E_g is the optical bandgap of the film determined from the normalized reduced absorption and PL spectra of films.

^b ΔV_{rad} is the voltage loss associated with radiative recombination.

^c $\Delta V_{non-rad}$ is the voltage loss associated with non-radiative recombination.

Comment 6: In general, when the emission device becomes saturation, EQE_{EL} should have a constant value. The authors' EQE_{EL} changes with the current. The authors seem to have calculated non-radiative losses using the values obtained at 1E-4 A. If calculated from 0.01 A, the EQE_{EL} values of the ternary and LBL-processed ternary OSCs are very close. The detailed values of the other energy loss terms will be changed. What is the basis for using the 1E-4 A for EQE_{EL} calculation?

Reply 6: At V_{oc} , the net current in the solar cell is zero, thus, the injection current density (J_{inj}) is equal to the photocurrent density (J_{ph}) generated in the solar cell. Therefore, J_{inj} used for

determining the EQE_{EL} must be equal to the J_{ph} , which is assumed to be equal to the short-circuit current density (J_{sc}) of the solar cell (Phy. Rev. B 2010, 81, 125204). The J_{sc} values of the solar cells studied in this work are about 25 mA cm^{-2} , and the active area of the device is 4 mm^2 , thus, we take the EQE_{EL} value at $\sim 10^{-4} \text{ A}$ for calculating the $\Delta V_{\text{non-rad}}$ (Figure R5).

Figure R5. EQE_{EL} spectra of OSCs at various injection current density.

Comment 7: In Supplementary Figure 21, there are 3 or 4 obvious peaks in the normalized EL curves, which have been rarely seen in other papers; what are the causes for this phenomenon?

Reply 7: Thanks for the comments. The multiple peaks in the normalized EL curves actually originate from the transition from the relaxed excited state to the different vibrational states in the ground state, as illustrated in Figure R6. There exists the high frequency vibrational modes for organic molecules, and the spacing between the vibrational energy levels is much larger than the thermal energy. Naturally, the electrons populate the lowest vibrational energy level in the ground state ($v=0$). After photon or electric excitation, excited states are formed, and the excited states quickly relax to the lowest vibrational energy state ($v'=0$), and then decay to the different vibrational energy states ($v=0, 1, 2$) in the ground state. The 0-0 transition leads to the emission of photons with energy E_{0-0} , which is related to the emission peak with the highest energy in the measured EL or PL spectrum, and the peaks with lower energy next to the 0-0 transition peak in the EL or PL spectrum, are associated with the 1-0 and 2-0 transitions (Sustainable Energy Fuels, 2018,2, 538-544).

Figure R6. Optical transitions depicted in an energy diagram with displaced potential wells for the ground state (GS) and excited state (ES), taking into account that the reaction coordinate remains invariant during the transition. Vertical blue arrows represent absorption and vertical red arrows represent emission (Sustainable Energy Fuels 2018, 2, 538-544).

Comment 8: Compared with BTP-eC9, L8 with a branched side-chain may possess less crystallinity, and why the introduction of L8 into the BTP-eC9-based active layer would improve the crystallinity of the acceptor phase. The lack of reasonable explanations discredited the results.

Reply 8: Thanks for the comments. The acceptor mixture that shows enhanced crystallinity has been reported by a few publications, which is not sole in our case (Adv. Mater. 2021, 33, 2007177; Small Sci. 2021, 2100092). The reasons that L8:BTP-eC9 mixture can be two fold. Firstly, L8 and BTP-eC9 have quite similar backbone structure that they have appropriate miscibility and similar crystalline structure, and probably forming eutectic mixture thus to enhance acceptor phase crystallinity. Secondly, we will need to consider the nucleation and growth procedure during crystal formation upon solvent removing. BTP-eC9 in processing solvent shows smaller solubility, which in solvent removal crash out earlier to form nuclei in larger quantity. This seeding process will lead to longer crystallization time and multi-centered crystal growth, which enhances mixture crystallinity.

Comment 9: As shown in the absorption spectra (Figure S1), when focusing on the absorption in 500-700 nm for the polymer donor, that in the LBL film is much lower than that in the BHJ one. It's hard to believe that the LBL and BHJ ternary devices delivered almost overlapped EQE responses (Fig. 1d).

Reply 9: Thanks for the comments. Figure S1b in the initial manuscript shows the normalized absorption of 120 nm-thick PM6:BTP-eC9 and PM6:BTP-eC9:L8-BO-F films. We remeasured the UV-vis absorption of the blend films. As shown in Figure R7, the absorption coefficients of ternary and LBL-processed ternary films are quite similar in the wavelength range of 500-700 nm, agreeing well with EQE spectra of the corresponding devices.

Figure R7. (a) UV-vis absorption spectra of 120 nm-thick PM6:BTP-eC9 and PM6:BTP-eC9:L8-BO-F films, and (b) EQE spectra of the corresponding devices.

Comment 10: In this article, there exist lots of deficiencies in language and grammar; for example, in Line 112-114, Page 5, the sentence "Our work not only highlights the role of……" seems incomplete; the indefinite article before FF should be "an" (Line 129-130, Page 6).

Reply 10: We thank the reviewer for pointing the mistakes. We carefully checked the whole manuscript and corrected the mistakes in the revised manuscript.

Reviewer #3 (Remarks to the Author):

In the ms, the authors reported the fabrication of thick-film OSCs using the ternary device strategy. A PCE of 17.31% was achieved for the 300 nm-thick device based on PM6:BTP-eC9:L8-BO-F blend. With 500 nm active layer, the device can deliver a PCE of 15.21%. The good devices results were thought to be ascribe to the increased exciton length for the ternary devices. In addition, the authors also gave some criteria for the active layer selection to fabrication thick film devices. Thick film devices with high efficiencies are indeed favorable for the large scale production of OSCs. This work can give some valuable insights for fabrication thick film OSCs active layers. It can be accepted after considering the following detailed comments.

Comment 1: As emphasized in the ms, the exciton diffusion length played the critical role for thick film device. What is the factors that determine the exciton diffusion length.

Reply 1: The common strategies to enhance singlet exciton diffusion include controlling the degree of crystallinity of materials, optimizing Forster energy transfer, and eliminating the exciton quenching defects (Energy Environ. Sci. 2015, 8, 1867-1888).

Comment 2: In the paragraph started from line 382, the authors gave some suggestions on the acceptor selection to obtain the enhanced exciton diffusion length. The role of the polymer donor is not mentioned. It should also play an important role on the exciton

diffusion length. More comments and explanation on it should be given.

Reply 2: We agree with the reviewer that the exciton diffusion in the domains of polymer donors is also critical for thick devices. In this study, we mainly focus on the study of a same polymer donor of PM6. Similar to most high-performance polymer donors, PM6 is a conjugated copolymer comprising electronic donating and withdrawing units. Due to aggregation, multiple forms of excited states are possibly involved in the dynamics so that it is difficult to evaluate the diffusion length using the power-dependent TA measurements. Nevertheless, in these systems, the long-range energy transfer is efficient (Nat. Mater. 2021, 20,378-384), which is likely to be responsible for the exciton diffusion in the polymer domains. While more-depth study is necessary towards a comprehensive understanding of the microscopic mechanism, it is likely a recommendable strategy to promote the exciton diffusion in polymer domains by optimizing the energy transfer via molecular design and morphology control.

Comment 3: Large area device at least 1cm^2 are suggested to give.

Reply 3: Large-scale device with 1 cm^2 active area was fabricated. *J-V* curve of the device and the corresponding EQE spectrum are shown in Figure R8. The large-area device yielded a PCE of 16.01%, with a J_{sc} of 28.38 mA cm^{-2} , a V_{oc} of 0.838 V, and an FF of 67.3%. Compared to the small-area device (0.04 cm^2), the large-area device suffers from the relatively low FF, mainly due to the limited conductivity of the ITO. The results and the corresponding descriptions of the large-area device were also added in the revised manuscript.

Figure R8. *J-V* curves of LBL-processed ternary OSC with 1 cm^2 active area, and the corresponding EQE spectrum. The inset is a photograph of the real large-area device.

Comment 4: The device stability measurement, at least some initial results are recommended to give for the thick film device.

Reply 4: Photostability of the thick devices was measured under continuous 1 sun illumination in air. As shown in Figure R9, 300 nm- and LBL-processed 300 nm-thick ternary

devices reached their T_{80} (the time required to reach 80% of initial performance) at 138, and 152 h, respectively. The more ordered molecular stacking, enhanced crystallinity, and reduced trap state density in LBL-processed thick films compared to the conventional thick films help increase the photostability of OSCs.

Figure R9. Photostability of 300 nm-thick ternary devices measured under continuous 1 sun illumination in air.

Reviewer comments, further review –

Reviewer #1 (Remarks to the Author):

The authors replied to all questions I had, incorporating novel data and analyses. I suggest to publish the manuscript as is.

Reviewer #2 (Remarks to the Author):

The quality of the revised manuscript has been greatly improved and I recommend accept it.

Reviewer #3 (Remarks to the Author):

The authors have carefully revised their work and given satisfied replies for reviewers' questions. I am glad to recommend it to be accepted.

Responses to the reports of the Reviewers

Reviewers' Comments:

Reviewer #1 (Remarks to the Author):

The authors replied to all questions I had, incorporating novel data and analyses. I suggest to publish the manuscript as is.

Reply: We thank the reviewer for the recommendation of our manuscript publishing in this journal.

Reviewer #2 (Remarks to the Author):

The quality of the revised manuscript has been greatly improved and I recommend accept it.

Reply: Thanks for the comments and the recommendation of our work for publication in this journal.

Reviewer #3 (Remarks to the Author):

The authors have carefully revised their work and given satisfied replies for reviewers' questions. I am glad to recommend it to be accepted.

Reply: We really appreciate the reviewer's positive evaluation and recommendation of our work.